# Evaluation of Antimicrobial Resistance in *Salmonella* Strains Isolated from Food, Animal and Human Samples between 2017 and 2021 in Southern Italy

**DOI:** 10.3390/microorganisms10040812

**Published:** 2022-04-13

**Authors:** Alessandra Alessiani, Elisa Goffredo, Maria Mancini, Gilda Occhiochiuso, Simona Faleo, Antonella Didonna, Rita Fischetto, Federica Suglia, Danila De Vito, Antonella Stallone, Luigi D’Attoli, Adelia Donatiello

**Affiliations:** 1Istituto Zooprofilattico Sperimentale della Puglia e della Basilicata, Via Manfredonia 20, 71121 Foggia, Italy; elisa.goffredo@izspb.it (E.G.); maria.mancini@izspb.it (M.M.); gilda.occhiochiuso@izspb.it (G.O.); simona.faleo@izspb.it (S.F.); antonella.didonna@izspb.it (A.D.); antonella.stallone@izspb.it (A.S.); luigi.dattoli@izspb.it (L.D.); adelia.donatiello@izspb.it (A.D.); 2Department of Paediatric Medicine, University Hospital Policlinic, PO Giovanni XXII, 70121 Bari, Italy; rfischetto@libero.it; 3Department of Basic Medical Sciences, Neurosciences and Sense Organs, University “Aldo Moro”, 70121 Bari, Italy; f.suglia2@studenti.uniba.it (F.S.); danila.devito@uniba.it (D.D.V.)

**Keywords:** *Salmonella*, Italy, *S.* Kentucky, *S.* Typhimurium, monophasic variant of Typhimurium, *S.* Derby, *S.* Infantis, antimicrobial resistance, MDR

## Abstract

*Salmonella enterica* is one of the most common causes of foodborne infection in the world, and the most common one in Italy. Italy collaborates with the other EU member states to survey the antimicrobial resistance of *Salmonella* on a large scale. This study on the situation in Apulia and Basilicata provides a more focused point of view on the territory, and anticipates the data reported in future Italian reports. Antimicrobial resistance was detected using the MIC detection method, with EUVSEC^®^ plates, on the strains collected between 2017 and 2021. The results of serotyping showed that *Salmonella* Infantis is the serovar that has increased the most over time in veterinary samples, while *Salmonella* Tyhimurium and its monophasic variant are the most isolated in human samples. The results of the antimicrobial resistance study comply with European data, showing high resistance to quinolones, tetracyclines, ampicillin and trimethoprim, and low resistance to colistin and cephems. The significant exception was that all strains were resistant to sulphametoxazole. The presence of MDRs, which was 85% in veterinary and 77.4% in human strains, often included critically important antibiotics, which is a sign that more study and action is needed to manage the use of antibiotics.

## 1. Introduction

*Salmonella enterica* is the second most commonly reported zoonotic agent in the European Union, after *Campylobacter* spp. Otherwise, *Salmonella* is the leading cause of foodborne disease in Italy, with 3268 cases reported in 2019, in contrast to the 1633 cases of campylobacteriosis [1]. A recent review showed that multi-drug-resistant non-typhoidal *Salmonella* may have more serious human health implications than those of pan-susceptible strains [1,2]. The Apulia and Basilicata regions cover almost 10% of the territory and comprise 7.6% of the population of Italy, and therefore are very important to the nation (Eurostat site http://dati-censimentipermanenti.istat.it/Index.aspx#21/12/2021 accessed on 21 December 2021). The Food Microbiology Unit of the Istituto Zooprofilattico Sperimentale di Puglia e Basilicata (IZSPB) collects *Salmonella* strains isolated in the territory of the two regions, according to national surveillance and monitoring plans, as well as for research and collaboration with universities and other research bodies, such as the Department of Basic Medical Sciences, Neurosciences and Sense Organs of Bari University, and the ENTER-NET local network, playing a fundamental role in epidemiological surveillance. The aim of this study was to evaluate the antimicrobial resistance of *Salmonella* serovars isolated from human, animal and food matrices in the Apulia and Basilicata regions between 2017 and 2021. In particular, this study of antimicrobial resistance focused on the five serovars mainly involved in human infections in Europe (8) isolated from food and animal sources: *S.* Infantis, *S.* Enteritidis, a monophasic variant of *S.* Typhimurium (MVST), *S.* Typhimurium and *S.* Derby [1]. In addition, *S.* Kentucky has been included in the study due to its growing relevance in laying hens in recent years [2].

## 2. Materials and Methods

### 2.1. Sample Collection

The strains were obtained from food and animal samples and routinely analyzed in the IZSPB laboratory as part of the official control activities according to European Regulations [3,4,5]. The food samples were meat (broiler, swine and cattle), mixed meat, mussels, milk (sheep), cheese (sheep) and feed. The animal samples were carcasses, feces, swabs from poultry (broiler and laying hens), cattle, sheep and pigs. Some strains were obtained from turtles and wild mammals from diagnostic or research activities. No animals were killed for this purpose.

Among the human origin strains collected in hospitals from symptomatic or asymptomatic patients and sent voluntarily from the hospital to the ENTER-NET local point, as written in the European Commission DG-SANCO Agreement N. SI2.326441, only strains that belonged to the six serotypes of interest were selected and sent to IZSPB.

### 2.2. Salmonella Isolation and Typing

The isolation of *Salmonella* strains from food, feed and animal samples was performed according to ISO 6579-1:2017 and ISO/TS 6579-2:2012 (microbiology of food and animal feeding stuffs: horizontal method for the detection of *Salmonella* spp.). The presence of *Salmonella* was confirmed using the appropriate biochemical miniaturized tests (API 20E^®^, Biomerieux, Lyon, France) and serological tests. Serotype identification of the isolated strains was carried out according to ISO/TR 6579-3:2014 (Microbiology of the food chain: horizontal method for the detection, enumeration and serotyping of *Salmonella*; Part 3: Guidelines for the serotyping of *Salmonella* spp.) using Statens Serum antisera (SSI Diagnostic, Hillerød, Denmark). One strain of *Salmonella* from each sample was preserved in a Microbank ™ vial (Pro-Lab Diagnostics, Richmond Hill, ON, Canada) at −20 °C.

### 2.3. Antimicrobial Susceptibility Testing

The *Salmonella* strain types, including Enteritidis, Infantis, Typhimurium, MVST, Kentucky and Derby, were characterized phenotypically by broth microdilution using Sensititre^®^ EUVSEC3^®^ and EUVSEC2^®^ plates (Termofisher Scientific, Paisley, UK). The antibiotics on the EUVSEC3^®^ plate were ampicillin (1–32 µg/mL), azithromycin (2–64 µg/mL), amikacin (4–128 µg/mL), gentamicin (0.5–16 µg/mL), tigecycline (0.25–8 µg/mL), ceftazidime (0.25–8 µg/mL), cefotaxime (0.25–4 µg/mL), colistin (1–16 µg/mL), nalidixic acid (4–64 µg/mL), tetracycline (2–32 µg/mL), trimethoprim (0.25–16 µg/mL), sulfamethoxazole (8–512 µg/mL), chloramphenicol (8–64 µg/mL), meropenem (0.03–16 µg/mL) and ciprofloxacin (0.015–8 µg/mL). The antibiotics on the EUVSEC2^®^ plate were cefoxitin (0.5–64 µg/mL), ertapenem (0.015–2 µg/mL), imipenem (0.12–16 µg/mL), cefotaxime (0.25–64 µg/mL), ceftazidime (0.25–12/8 µg/mL), cefepime (0.06–32 µg/mL), cefotaxime/clavulanic acid (0.06/4–64/4 µg/mL), ceftazidime/clavulanic acid (0.12/4–128/4 µg/mL) and temocillin (0.5–128 µg/mL). The plates contained the antimicrobials specified in the Commission Implementing Decision (EU) 2020/1729 [6]. The definition of sensibility or resistance was based on the Epidemiological Cut-Off value (ECOFF) shown in a previously indicated decision, except for azithromycin, for which the sensibility or resistance values reported in the CLSI 2021 document M100 were used, due to the lack of specific indications in the aforementioned legislation [7]. The quality control of the batch was performed with *E. coli* ATCC^®^ 25922. The strains were referred to as multi-drug-resistant (MDR) when they simultaneously showed resistance to at least three different classes of antibiotics.

The antimicrobial test was carried out on colonies grown on non-selective agar medium (Tryptic Soy Agar, VWR, Milan, Italy) incubated at 36 ± 1 °C overnight, touched with a sterile swab, and transferred to sterile saline solution until they reached 0.5 McFarland turbidity. Then, 0.1 mL of each suspension was inoculated in 9.9 mL of cation-adjusted Mueller Hinton Broth (Becton Dickinson, Milan, Italy), and 50 µL of this suspension was added into the wells of the EUVSEC3^®^ plates. The EUVSEC2^®^ plates were used only to check the cephalosporine and carbapenems resistance when it occurred. The plates were incubated aerobically at 36 ± 1 °C for 24 h before the reading.

## 3. Results

### 3.1. Salmonella Strains Collected and Typed

The 1050 strains collected in IZSPB between 2017 and 2021 from food and animal (indicated below as FA) matrices comprised 337 *S.* Infantis samples (32.1%), 118 *S.* Kentucky samples (11.2%), 64 monophasic variants of *Salmonella* Typhimurium (MVST) (6.1%), 55 *S.* Typhimurium samples (5.2%), 40 *S.* Derby samples (3.8%), 30 *S.* London samples (2.9%), 29 *S.* Bredeney samples (2.8%), 26 *S.* Abony samples (2.5%), 25 *S.* Kaseny samples (2.4%), 22 *S.* Muenster samples (2.1%), 19 *S.* Agona samples (1.2%), 18 *S.* Anatum samples (1.7%), 17 *S.* Enteritidis samples (1.6%), 16 *S.* Rissen samples (1.5%), 13 S. Newport samples (1.2%) and 10 *S.* Branderup samples (1%). The remaining 275 strains belonged to 45 other serotypes with less than 10 strains per serotype.

*Salmonella* Enteritidis, *S.* Derby, *S.* Kentucky, *S.* Infantis, *S.* Typhimurium and its monophasic variant (MVST) represented approximately 60% of the typed strains. The distribution per year of these strains is shown in Figure 1.

The 146 *Salmonella* strains of human origin (indicated below as H) collected between 2017 and 2021 by the ENTER-NET local network (Department of Basic Medical Sciences, Neurosciences and Sense Organs of Bari University) and sent to IZSPB comprised 5 *S.* Infantis samples (3.4%), 5 *S.* Derby samples (3.4%), 29 *S.* Enteritidis samples (19.9%), 1 *S.* Kentucky samples (0.7%), 39 monophasic variants of *Salmonella* Typhimurium (MVST) (26.7%), and 68 *S.* Typhimurium samples (46.6%). Figure 2 shows the distribution per year of these strains.

### 3.2. Distribution of Sample

The food, feed and animal matrices positive for *Salmonella* spp. sampled within the scope of the National Control Plan for foods of animal origin comprised 49.6% of the sample (Reg. (EU) 2017/625). The *Salmonella* National Control Program in poultry population comprised 27.4% (Reg. (EC) 2160/2003); screening, research cooperation and other unspecified reasons comprised 14.1%; outbreak investigation comprised 3.4%; and the National Control Plan for feed comprised 3.2% (Reg. (EU) 2017/625) [3,4]. The *Salmonella* strains were detected principally in food and animal samples (56.1% and 41.3%, respectively) and to a small extent in feed (2.6%). The main source of *Salmonella* in food was chicken meat, with about 27% of the total isolates (1050 strains), followed by pork meat (10.3%), mixed meat preparations (5.6%), clams (3.7%), and bovine meat (3.2%). Regarding animal carcasses and swabs, *Salmonella* was isolated primarily from laying hens (13.4% on 1050 strains), followed by broilers (10.8%), bovine (4.5%), and turtles (4%). Food and animal strains originated from the following provinces of the Apulia and Basilicata regions: Bari (22.3%), Foggia (21.1%), Matera (15.9%), Brindisi (9.2%), Potenza (8.8%), Barletta-Andria-Trani (8.2%), Lecce (8.2%) and Taranto (6.4%).

Of the 146 human strains, 67 originated from children aged 0–6 years old (45.9%), 16 from children aged 7–18 years old (11%), 11 from adults aged 18–60 years old (7.5%) and 11 from adults aged 60–90 years old (7.5%), and the age was unknown for the remaining 39 samples. The human samples originated from the following geographical areas: Bari (63%), Foggia (8.2%), Matera (1.4%), Brindisi (26%), Barletta-Andria-Trani (1.4%) and territories outside Apulia and Basilicata regions (1.4%).

### 3.3. Antimicrobial Susceptibility Testing

Regarding *Salmonella* strains isolated from food and animal matrices, the antimicrobial susceptibility test was performed only on those belonging to the six serovars of interest (206): 100 *S.* Infantis; 35 *S.* Kentucky; 27 MVST; 26 *S.* Typhimurium; 13 *S.* Derby; and 5 *S.* Enteritidis.

All 146 human strains sent to IZSPB were examined.

Among the FA strains, the most resistant serotype to all evaluated antimicrobial classes was *S.* Infantis, while for H strains, it was *S.* Typhimurium. 

All *Salmonella* strains were resistant to sulfamethoxazole. None were resistant to meropenem.

*Salmonella* FA strains showed antimicrobial resistance, mainly to nalidixic acid (63.6%), ciprofloxacin and tetracycline (60.7% each), trimethoprim (42.7%), ampicillin (41.3%), cefotaxime (17.5%) and ceftazidime (14.1%), followed by other antimicrobial classes with resistance percentages less than 10% (Table 1.). 

*Salmonella* H strain resistance was detected mainly for ampicillin (57.5%), tetracycline (50.7%), azithromycin (36.3%), chloramphenicol (9.6%), ciprofloxacin (8.9%) and nalidixic acid (8.2%) followed by other antimicrobial classes with resistance percentages of less than 8%. 

Table 1 shows the resistance percentage of the total strains, distinguished with respect to H and FA.

Among the FA strains, *S.* Infantis displayed resistance mainly to tetracycline (73%), followed by nalidixic acid and ciprofloxacin (70% and 69%, respectively), trimethoprim and ampicillin (55% and 44%, respectively) and cephems (19–23%). Ciprofloxacin and nalidixic acid resistance were predominant in the *S.* Derby strains (76.9% and 69.2%, respectively), followed by trimethoprim (61.5%), ampicillin (46.2%), ceftazidime and cefotaxime (30.8% each). The MVST strains were mainly resistant to tetracycline (70.4%) and ampicillin (63%). *S.* Typhimurium strains were resistant to nalidixic acid (50%), tetracyclines and ampicillin (42.3% each) and ciprofloxacin (38.5%). Table 2 shows all resistance percentages of the FA strains.

For the H strains, *S.* Infantis showed simultaneous resistance to ampicillin, ciprofloxacin, trimethoprim and azithromycin (85.7%); they were also resistant to both nalidixic acid and tetracycline (71.4%), to cephems (57.1%) and to tigecycline (28.6%). The highest levels of resistance were observed in *S.* Derby and *S.* Enteritidis to azithromycin (66.7% and 14.8%, respectively). The MVST strains revealed a resistant phenotype mainly to ampicillin (63.6%), followed by tetracycline and azithromycin (57.6% and 42.4%, respectively), as well as *S.* Typhimurium, whose resistance percentages were 75%, 65.3% and 34.7%, respectively. Table 3 shows all the resistances.

Resistance to two or more antimicrobials was detected in 175 out of the 206 FA strains (85%) and in 113 out of the 146 H strains (77.4%).

The most common MDR pattern found in all FA *Salmonella* strains was CIP-NAL-TET-TRI-SUL, which is associated with other antimicrobials, with the exception of MVST, which did not show as much resistance to quinolones.

*S.* Infantis showed the main MDR pattern detected in FA strains: CIP-NAL-TET-TRI-SUL-AMP-FOT- TAZ in 14 strains; and CIP-NAL-TET-TRI-SUL in 9 strains. The most common MDR pattern in H strains was AMP-TET-SUL, which was associated with other antimicrobials. The main MDR in H strains was AMP-TET-SUL-AZM, detected in 11 *S.* Typhimurium and in 8 MVST strains. *S.* Infantis with this kind of multidrug resistance was included in a wider multidrug resistance pattern, comprising up to 11 types of resistance (AMP-CIP-AZM-TIG-FOT-TAZ-COL-NAL-TET-TRI-SUL), and it was also detected in *S.* Derby and *S*. Enteritidis, which are associated with other antimicrobials.

## 4. Discussion

The extreme diffusion of *Salmonella* spp. and its ability to infect humans from animal, food and environmental sources makes this bacterium very important in health surveillance systems. Several European regulations provide rules regarding the management of *Salmonella* and the surveillance of antibiotic resistance; thus, this contributes to producing a large amount of data related to these topics. In Europe, the trend of human salmonellosis has been stable over the last five years, while *Salmonella* positivity has increased significantly in laying hens and breeding flocks [8]. The top five *Salmonella* serovars for human infections from food–animal sources are *S.* Infantis, *S.* Enteritidis, a monophasic variant of *Salmonella* Typhimurium, *S.* Typhimurium and *S.* Derby [8]. The vast majority of the salmonellosis foodborne outbreaks are caused by *S.* Enteritidis, and the most implicated food vehicles are eggs and egg products [1,8]. The relevance of *S.* Kentucky is growing (2) and, in our opinion, even if it has little impact on human cases, it could be a source of antibiotic resistance to be considered in relation to European data [1].

In the last five years, *S.* Infantis from FA samples has accounted for more than 30% of the total strains isolated in the Apulia and Basilicata regions. The prevalence value has increased over time: it was detected only 12 times in 2017, but 106 samples were positive in 2021, as shown in Figure 1. In particular, *S.* Infantis was detected in 94% of broiler samples and in only 6% of other sources. This result is in agreement with the European data, where *S.* Infantis was found to be the most frequently isolated serovar in broiler flocks and the fourth most frequent one in breeding flocks and laying hens in the member states [1]. The significant increase in the prevalence of *S.* Infantis seems to be linked to the presence of an ESBL-producing multi-drug-resistant clone harboring a conjugative pESI-like meg plasmid, which has been detected since 2014 [2,9,10]. The characteristic resistance pattern of pESI-like meg *S.* Infantis is towards tetracyclines, trimethoprim, sulphonamides and aminoglycosides associated with fluoroquinolone and macrolide resistance [10]. A stable population of this kind of *S.* Infantis has been recorded in Italy [2,9]. Our data seem to confirm this, since a lot of *S.* Infantis strains isolated from both human and veterinary samples displayed simultaneous resistance to sulphonamides, tetracyclines and trimethoprim, which are sometimes associated with fluoroquinolones, aminoglycosides or beta-lactams. The MDR to HpCIAs is particularly interesting, and signals that there is still much work to be completed on the use of these antibiotics, as well as on EsBL resistance. In 23 FA and 3 H strains, there was resistance to one of the tested cephems, showing a worrying incidence of this kind of resistance, as reported by many authors [10,11]. Colistin resistance was very low, being detected in only two FA strains and one H strain; this result is in agreement with the European data [8]. Both EsBL and colistin resistance were associated with MDR complexes, and reached 11 antimicrobial resistances at the same time.

*S.* Enteritidis emerged as a global epidemic in the 1980s and persists in most countries as the leading cause of human salmonellosis due to its ability to colonize reproductive systems in poultry and contaminate eggs [2]. It is the second most common serovar in laying hens, but European data show that is very infrequent in broilers [2,8]. The distribution of *S.* Enteritidis detected in our FA samples was slightly different, but the low number of strains examined does not allow for too detailed considerations. There were 27 strains detected in H samples out of 146, and there were two colistin-resistant strains, which is consistent with a previous study on ENTER-NET data [12]. Resistance to azithromycin was the most frequent one in *S.* Enteritidis strains of human origin, followed by resistance to quinolones and fluoroquinolones. Conversely, this resistance was nonexistent in strains of veterinary origin, which showed resistance mostly to quinolone and fluoroquinolone. Some H and FA strains were MDR, showing resistance to colistin and cefotaxime/ceftazidime, respectively, and always associated with other antimicrobial agents. In these MDR strains, there were many HpCIAs and up to nine types of resistance at the same time, making them particularly dangerous for human health. It should be noted that there was a strain of animal origin and a human one with the same MDR pattern (AMP-CIP-TAZ-FOT-NAL-TET-TRI-SUL), except for the resistance to azithromycin, which was added to the human strain. This kind of MDR pattern could be of great concern for humans; if humans become infected with the strain from food or animals they can provide it an additional chance to acquire antimicrobial resistance, worsening the features of this strain. Furthermore, given the great invasion capacity of *S.* Enteritidis, the fact that three out of five strains of veterinary origin were MDR is of great concern for public health.

*S.* Typhimurium is the second most common serovar in humans with epidemic potential in animal and human populations. This kind of *Salmonella* often evolves towards multiple types of antimicrobial resistance, such as the DT104 strain, which has enhanced virulence in humans and animals [2,13,14]. The monophasic variant *of S.* Typhimurium (MVST) strains emerged in pig populations in 1980 and rapidly spread, reaching a prevalence value of 8.3% in samples of animal origin and 14% in samples of human origin in 2008 [9]. MVST and *S.* Typhimurium were the most frequent serovars isolated from humans in Italy, representing 35% and 10.2%, respectively, of all reported isolates between 2016 and 2018 [2,15]. Nowadays, they belong to the top five serovars in humans and animals [8]. The presence of integrons, invasion (*invA*) and virulence (*spv*, *Salmonella* plasmid virulence) genes plays a fundamental role in the spread of this kind of *Salmonella*, whose main source is presently from pigs [13,14]. Furthermore, some studies have reported the presence of *S.* Typhimurium and MVST in wildlife mammals in southern and northern Italy [16,17]. One of the latest major outbreaks of MVST detected in Italy occurred in the Abruzzo region between 2013 and 2014, and involved 30 people; it was caused by a particular MDR profile [15], to which more attention should be paid because of its ability to persist in the environment and its infectious capacity. Overall, in our study, 107 H and 119 FA strains of *S.* Typhimurium and its monophasic variant were analyzed. The most common instances of resistance in human strains were to AMP, TET, AZM and SUL. Chloramphenicol resistance seems to occur frequently. Streptomycin was not tested in our panel; therefore, it is not possible to include these strains in the ASSuT or ACSSuT patterns. An assessment of the ASSuT or ACSSuT patterns will be conducted by molecular biology methods in our future studies [18,19].

*S.* Kentucky, accounting for about 40% of all identified isolates each year in laying hens in Italy, was the most frequently detected serovar by far. Conversely, it was detected in humans at a low frequency (0.2%) [2]. Our study reveals that the presence of *S.* Kentucky increased from 2017 to 2021. In particular, 64% of *S.* Kentucky originated from laying hens and 23% from broilers; however, in our opinion, it is interesting to note that 12.7% of strains were collected from other sources (swine, bovine, turtles and wild birds), as well as the detection of one human strain in 2021. *S.* Kentucky is an emergent strain, and the sequence type ST198 has rapidly spread in broiler populations in many EU and non-EU countries [8,20,21,22]. An in-depth study will be conducted on the STs in our population in order to assess whether they belong to the clones circulating in Europe. The H strain of *S.* Kentucky was sensible to all antimicrobial agents. The occurrence of MDR in FA *S.* Kentucky strains was not so abundant, but among MDR strains there were four strains resistant to cephems and one strain resistant to colistin. The MDR strains comprised 28.6% of the total. These data contrast with those shown by the EFSA, where MDR was reported in 73.7% of *S.* Kentucky isolated from human cases [23]. The resistance to quinolones and fluoroquinolones has been confirmed as the most frequent occurrence, in agreement with the EFSA report; in fact, in our study, 77.1% of *S.* Kentucky strains were resistant to ciprofloxacin, and the EFSA reports 82.1% associated predominantly with poultry [23].

According to the ENTER-NET network data, *S.* Derby represented 3% of all reported isolates between 2016 and 2018 [2]; in 2019, their ESBL production was highlighted [23]. In our study, we can assume that, based on the evaluation of MICs, the presence of ESBL strains mean that *S.* Derby should be one of the most supervised strains, even though it did not have a prevalence comparable to the other *Salmonella* serovars in the present study. *S.* Derby was detected in 40 FA and 6 H samples. Eleven FA strains were MDR, and four of these were resistant to CIP-NAL-TET-TRI-SUL. Among the H strains, two were MDR, and one of these had cephem resistance, which was confirmed when testing the same strains with the EUVEC2 ^®^ plate but not with the molecular detection of specific genes.

In our study, of the human strains, resistance was most observed to ampicillin, sulfamethoxazole and tetracyclines. The level of resistance to cefotaxime and ceftazidime was low (0.03%), and none of the strains were resistant to meropenem.

In FA strains, the resistance percentage value versus ampicillin was lower than the value for ciprofloxacin, nalidixic acid, tetracyclines and sulfamethoxazole, but it still remained quite high. The resistance to cefotaxime and ceftazidime was at a medium level of 17.5%, higher than that found in 2019 according to the monitoring plans of the EU member states [8,23]. Sulfamethoxazole and tetracyclines are categorized as highly important antimicrobials by the World Health Organization, and despite the invitation to moderate or exclude their use, the resistance remains high, as shown by our survey; the same can be said for the use of quinolones and fluoroquinolones. However, the resistance to azithromycin, tigecycline and colistin was low, as shown in the European data [1].

In conclusion, our data, which were limited to the Apulia and Basilicata regions, were obtained from the antibiotic panel of the EU Decision 2020/1729, with the aim of making our data comparable with European data. It can be argued that our data agree with other European data, although they differ from other Italian studies [6,16,17].

The resistance to sulfamethoxazole shown by 100% of the strains is certainly noteworthy, especially since our data are more similar to those registered in Russia and China rather than Europe [23,24,25]. We suspect that this high prevalence of resistance to sulfamethoxazole is linked to the ease of spreading type 1 integron associated with the presence of the aadA2 gene, which perhaps will take over in the next few years. This topic will certainly be a starting point for further insight into the molecular biology of *Salmonella* in future studies, for example with the use of whole genome sequencing.

## Figures and Tables

**Figure 1 microorganisms-10-00812-f001:**
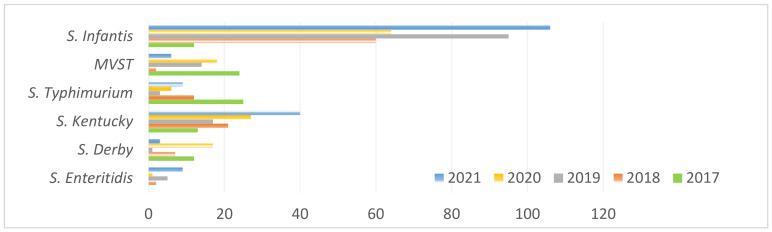
Distribution of *S.* Infantis, MVST, *S.* Typhimurium, *S.* Kentucky, *S.* Derby and *S*. Enteritidis per year. FA strains.

**Figure 2 microorganisms-10-00812-f002:**
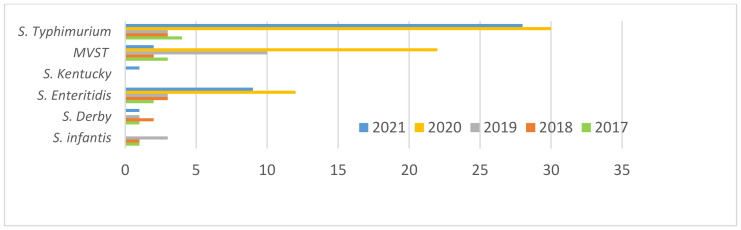
Distribution of *S.* Infantis, MVST, *S.* Typhimurium, *S.* Kentucky, *S.* Derby and *S*. Enteritidis per year. H strains.

**Table 1 microorganisms-10-00812-t001:** Resistance percentage of the total strains to single antimicrobial agents. In bold are the highest priority critically important antimicrobials.

Antimicrobial Agents	Human-Resistant Strains/TOT	Food–Animal-Resistant Strains/TOT
Sulfamethoxazole (SUL)	100.0%	100.0%
**Nalidixic Acid (NAL)**	**8.2%**	**63.6%**
**Ciprofloxacin (CIP)**	**8.9%**	**60.7%**
Tetracycline (TET)	50.7%	60.7%
Trimethoprim (TRI)	6.2%	42.7%
Ampicillin (AMP)	57.5%	41.3%
**Cefotaxime (FOT)**	**4.1%**	**17.5%**
**Ceftazidime (TAZ)**	**3.4%**	**14.1%**
Tigecycline (TGC)	2.7%	7.8%
**Azithromycin (AZM)**	**36.3%**	**7.3%**
Chloramphenicol (CLO)	9.6%	6.3%
Amikacin (AMI)	3.4%	4.4%
**Colistin (COL)**	**2.7%**	**2.9%**
Gentamicin (GEN)	0.0%	1.5%
Meropenem (MERO)	0.0%	0.0%

**Table 2 microorganisms-10-00812-t002:** Resistance to single antimicrobial agents for the serotypes of FA strains. In bold are the highest priority critically important antimicrobials.

Antimicrobial Agents	*S.* Infantis	*S.* Derby	*S.* Enteritidis	*S.* Kentucky	VMST	*S.* Typhimurium
Ampicillin	44.0%	46.2%	40.0%	14.3%	63.0%	42.3%
Meropenem	0.0%	0.0%	0.0%	0.0%	0.0%	0.0%
**Ciprofloxacin**	69.0%	76.9%	80.0%	77.1%	18.5%	38.5%
Azithromycin	9.0%	0.0%	0.0%	5.7%	3.7%	11.5%
Amikacin	5.0%	15.4%	0.0%	2.9%	3.7%	0.0%
Gentamicin	0.0%	7.7%	0.0%	0.0%	7.4%	0.0%
Tigecycline	11.0%	7.7%	0.0%	2.9%	11.1%	0.0%
**Ceftazidime**	19.0%	30.8%	20.0%	5.7%	3.7%	7.7%
**Cefotaxime**	23.0%	30.8%	40.0%	11.4%	0.0%	11.5%
Chloramphenicol	5.0%	0.0%	0.0%	2.9%	11.1%	15.4%
**Colistin**	2.0%	7.7%	0.0%	2.9%	0.0%	7.7%
**Nalidixic Acid**	70.0%	69.23%	80.0%	85.7%	18.5%	50.0%
Tetracycline	73.0%	53.9%	60.0%	34.3%	70.4%	42.3%
Trimethoprim	55.0%	61.5%	60.0%	25.7%	29.6%	19.2%
Sulfamethoxazole	100.0%	100.0%	100.0%	100.0%	100.0%	100.0%

**Table 3 microorganisms-10-00812-t003:** Resistance to single antimicrobial agents in the serotypes of H strains. In bold are the highest priority critically important antimicrobials.

Antimicrobial Agents	*S.* Infantis	*S.* Derby	*S.* Enteritidis	*S.* Kentucky	MV*S*T	*S.* Typhimurium
Ampicillin	85.7%	33.3%	3.7%	0.0%	63.6%	75.0%
Meropenem	0.0%	0.0%	0.0%	0.0%	0.0%	0.0%
**Ciprofloxacin**	85.7%	16.7%	0.0%	0.0%	6.1%	5.6%
**Azithromycin**	85.7%	66.7%	14.8%	0.0%	42.4%	34.7%
Amikacin	14.3%	16.7%	3.7%	0.0%	3.0%	1.4%
Gentamicin	0.0%	0.0%	0.0%	0.0%	0.0%	0.0%
Tigecycline	28.6%	0.0%	3.7%	0.0%	0.0%	1.4%
**Ceftazidime**	57.1%	16.7%	0.0%	0.0%	0.0%	0.0%
**Cefotaxime**	57.1%	16.7%	0.0%	0.0%	0.0%	1.4%
Chloramphenicol	14.3%	16.7%	7.4%	0.0%	6.1%	11.1%
**Colistin**	14.3%	0.0%	7.4%	0.0%	0.0%	1.4%
**Nalidixic Acid**	71.4%	0.0%	7.4%	0.0%	9.1%	2.8%
Tetracycline	71.4%	33.3%	3.7%	0.0%	57.6%	65.3%
Trimethoprim	85.7%	0.0%	0.0%	0.0%	3.0%	2.8%
Sulfamethoxazole	100.0%	100.0%	100.0%	100.0%	100.0%	100.0%

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
