# Peer review of "Evaluation of Antimicrobial Resistance in Salmonella Strains Isolated from Food, Animal and Human Samples between 2017 and 2021 in Southern Italy"

_microorganisms, 2022, doi:10.3390/microorganisms10040812_

Round 1

Reviewer 1 Report

The authors investigated the antimicrobial resistance of Salmonella strains isolated from human, animal and food samples in Apulia and Basilicata regions between 2017-2021. To do so, they sampled food, animal and humans and they performed antimicrobial susceptibility testing to the isolated bacteria.

The scope of this paper is in line with those of Microorganisms and the analysis performed are scientifically sound. However, I would recommend that the authors carefully revise the points/suggestions below:

  1. Title: “on salmonella strains” should be “in Salmonella strains” and “Salmonella” should be written in italic.
  2. Abstract: Line 20-21: Infantis and S. Tyhimurium should be written in full (because it is first use).
  3. Line 24: “Sulphametoxazolo” should be “sulfamethoxazole”.
  4. Line 28-29: species names should be in Italic.
  5. Line 54: “Samples collection” should be “Sample collection”.
  6. Line 55: “the food and animal samples collection was obtained” should be corrected as follow: “the food and animal samples were obtained” or “The collection of samples from food and animal was obtained”.
  7. Lines 65, 69, 70, 206: “Salmonella” should be in italic and please revise along the manuscript text.
  8. Lines 92-96: Add percentages in front of the numbers.
  9. line 99: Delete “kind” and correct as follow: “of these strains”.
  10. Figure 1: in the title and the figure’s axis, species names should be in italic.
  11. Line 108: Delete “kind” and correct as follow: “of these strains”.
  12. Line 126: Please delete “etc.” or mention the remaining.
  13. Line 132: Please correct as follow: “were originated from ..”
  14. Line 129: “on” should be “for”.
  15. Line 114: Subtitle should be “territorial distribution of sample” Please correct.
  16. Lines 138-139: Please use abbreviations for species names ( infantis, S. kentucky etc) along the text since it is already written in full in the abstract and the introduction.
  17. Line 146: ciprofloxacin and tetracycline (60,68%) each?
  18. Line 148: Add “Table 1” between brackets.
  19. Line 164: Please use abbreviations for species names.
  20. Table 2 and Table 3: species names should be in italic (Genus and species: Names should always be italicized).
  21. Line 172: “it was also resistant both to nalidixic acid and tetracycline (71,43%)” should be “they were also both resistant to nalidixic acid and tetracycline (71,43%)”.
  22. Line 174: “observed on” should be “observed in”.
  23. Line 186: “didn’t” should be “did not”.
  24. Line 188: species name in abbreviation. “Two main MDRs detected” should be “main MDR patterns detected”.
  25. line 192: Add “pattern” to be “MDR pattern”.
  26. In “Materials and Methods” section:
  • Information about sampling sites is missing (animal and human sampling site).
  • The food samples: please specify.
  • Animal species are missing (which animal species were studied).
  • The set of antibiotics tested is missing (which antibiotics were studied).
  1. In “Results” section:
  • FA: for what reason the authors joined together the food and animal, why they do not separate the 2 results? For the result clarity, I would suggest presenting results of food and animal separately also.
  1. Please revise along the text: species should be in italic including Figures and Tables.

      In its first use within a manuscript, the genus is always written in full. In subsequent uses, the genus can be abbreviated using the first initial. Please apply in the text.

  1. The antibiotics that showed high resistance in this study are they used in in human clinic and animal setting in Italy?? Information regarding the use of antibiotics in the Italy in both hospital and veterinary settings are missing.
  2. Do authors perform some statistical analysis?
  3. Conclusion of the study is missing
  4. I suggest that authors should focus more on the writing and make sure to check spelling and grammar and fix any typos or unclear sentences.
  5. Discussion: I would recommend softening a bit the discussion and simplifying.
  6. May it be interesting if the authors try to perform further studies such as whole genome sequencing and investigate the gene content and transfer between animal, food, and human. This may provide a support to the findings of this study.

Author Response

Dear Reviewer 1,

thank you very much to your suggestions and corrections. We made some modification to grant your requests and we insert below some responses to your comments to better explain our point of view.

We have made all the correction listed between 1-26-28 point and we hope have to correct any typo error, whit the help of an English revision process.

27) We have considered the strains from animal and food like unique group to comment more fluently the article being infections carried by foods of animal origin mainly. The principal reason of difference from food and animal was the regulation of their sampling.

29) The antibiotic chosen were indicated on EU Decision 2020/1729 to monitoring and reporting of antimicrobial resistance in zoonotic and commensal bacteria and we use this panel whit the aim of making our data comparable with the European ones about surveillance. Of course, they are used on human and veterinary medicine, but the use is regulated from specific lows and to insert comment about these it would be off topic, in our opinion.

30) None.

31) The paragraph “conclusion” is not necessary on the Microorganism form, but I have modified the last paragraph to emphasize our final comment.

32) Done.

33) Some phrases were modified.

34) It will be our future project about these strains, because right now it would be off topic.

Best regards,

Alessandra Alessiani

Reviewer 2 Report

Dear authors,

I have comments and corrections marked in the text for you. Please make them.

Author Response

Dear Reviewer 2,

thank you very much to your suggestions. I have made all the corrections required.

Best regards,

Alessandra Alessiani

Reviewer 3 Report

Thank you authors for the interesting findings. The study summarised antimicrobial resistance in selected Salmonella serovars isolated from food chain and humans over a period.
Some suggestions as below for consideration.

  • Line 54 Samples collection ; what is the basis of testing asymptomatic patients? what types of samples e.g. stools? and whether consent has been taken though it is noted as voluntary. Please also include ethic consideration statement, since there seems to be human subjects involved.
  • Line 73 Antimicrobial Susceptibility Testing ; please indicate how MDR is being defined/categorised
  • Throughout the manuscript ; is there a need of 2-decimals for percentage figures? suggest to round up to 1-decimal, else the numbers with too many decimals can be quite distracting for readers diluting the key messages.
  • Line 100 and 110 - Figure 1 and 2 can be summarised into a table (e.g. 3 columns) showing serovars, number of isolates from food-animals, and number of isolates from humans by years.
  • Line 203 ; top five Salmonella serovars for human infections from food-animal sources ; how do we know if the infections are definitive from food-animal sources? is this referred to the findings of this study or literature? if it is the former, this study does not provide any source attribution information or human cases' food consumption history e.g. outbreak investigations. if it is the latter, suggest a citation.
  • Results ; as the study applied MIC method for AST, MIC info would be value-added to report in the manuscripts.
  • Line 297-298 ; discussion on ESBL production relating to S. Derby. did the authors conduct any tests to confirm ESBL-production in Salmonella isolates including in this study? else, how do we relate to this discussion point on ESBL in line 298?

Author Response

Dear Reviewer 3,

thank you very much to your suggestions. We have made some modification to grant your requests and we insert below some responses to your comments to better explain our point of view.

Line 54) Sample collection. The human samples were collected from hospitals based on their procedure that were out of topic for our article. The Salmonella strains were sent voluntary to the ENTER NET local point based on decision of European Commission (DG-SANCO Agreement N. SI2.326441 (2001)). We have modified the paragraph to better explain this flow. There is not necessary to have other any ethic consideration statement.

Line 73) Done.

Throughout the manuscript:

We have corrected the percentage whit 1-decimal.

We did not consider it appropriate to modify figures 1 and 2 in a table, because we felt that in this way it was more immediate to understand the numerical variation that has occurred over time.

Line 203) We have modified to explaining it better.

Results: Your suggestion was very appropriate, but we think it would make the article too heavy, diverting attention from the focus. In fact, certainly there are various degrees of MIC about resistance and sensitivity, but they will be the subject of other works, more focused on this kind of information.

Line 297-298.

The study about ESBL production will be performed in future study, we just can suppose it was based on MIC. We have better explained the statement.

Best regards,

Alessandra Alessiani